# Protective Effects of Chestnut (*Castanea crenata*) Inner Shell Extract in Macrophage-Driven Emphysematous Lesion Induced by Cigarette Smoke Condensate

**DOI:** 10.3390/nu15020253

**Published:** 2023-01-04

**Authors:** Ji-Soo Jeong, Jeong-Won Kim, Jin-Hwa Kim, Chang-Yeop Kim, Je-Won Ko, Tae-Won Kim

**Affiliations:** BK21 FOUR Program, College of Veterinary Medicine, Chungnam National University, 99 Daehak-ro, Daejeon 34134, Republic of Korea

**Keywords:** chestnut inner shell, cigarette smoke condensate, emphysema, matrix metalloproteinase

## Abstract

Chestnut (*Castanea crenata*) inner shell extract (CIE), a curative herb in Korea, has diverse pharmacological effects against various diseases including pulmonary fibrosis, asthma, and chronic obstructive pulmonary disease (COPD). However, its molecular mechanisms of anti-emphysematous effects are still not fully elucidated. In the present study, we elucidate the efficacy of CIE against emphysematous lesion progression in a cigarette smoke condensate (CSC)-instilled mice and CSC-stimulated H292 cell line. The mice are administered CSC via intranasal instillation at 7-day intervals for 1 month after 1 week of pretreatment with CIE. CIE (100 or 300 mg/kg) is administered by oral gavage for 1 month. CIE decreased the macrophage count in bronchoalveolar lavage fluid and the severity of emphysematous lesions in lung tissue. Additionally, CIE suppressed the phosphatidylinositol 3-kinase/protein kinase B/nuclear factor kappa B signal pathway and thereby downregulated matrix metalloprotease-9 expression, which was confirmed in CSC-stimulated H292 cells. Thus, CIE effectively inhibited CSC-induced macrophage-driven emphysema progression in airways; this inhibition was associated with the suppression of protease–antiprotease imbalance. Our results propose that CIE has the potential for the alleviation of COPD.

## 1. Introduction

Chronic obstructive pulmonary disease (COPD) is a lung disease triggered by sustained airflow limitation for which prevention and treatment are important [1]. Such a limitation is associated with airway and/or alveolar abnormalities caused by substantial exposure to toxic particles or gases [2]. Various genetic and environmental factors contribute to COPD; among them, the main pathological factor is cigarette smoke (CS) [3]. CS is an abundant source of oxidants, which recruit macrophages into the lungs [4,5]. Among the macrophage-related mechanisms of emphysema progression, protease–antiprotease imbalance is one of the main contributors to the development of emphysema [6,7,8]. Macrophage infiltration into the lungs causes COPD by inducing the release of tissue-destructive proteases, including matrix metalloproteases (MMPs) [9]. If the released proteolytic enzymes are not completely suppressed by anti-proteases, they would eventually degrade the alveolar wall matrix and cause emphysema [10]. The levels of MMP-9 increased in a CS-induced lung injury mice model and in clinical trials for smokers [11,12]. These results strongly indicate that proteases play a role in emphysema progression. Therefore, the regulation of protease–antiprotease imbalance is considered an important strategy for reducing the occurrence of emphysema in CS-induced COPD.

In the past few years, natural antioxidants in fruits, vegetables, plants, and traditional medicinal herbs have attracted attention [13]. In particular, interest in polyphenolic antioxidants has increased; natural polyphenols can protect against injurious free radicals, regulate immune reaction, and alleviate the symptom of serious illness such as cancer and inflammation [14,15]. The inner shells of chestnuts, which are used as a medicinal herb in Korea, exert various pharmacological effects derived from their large contents of polyphenolic components, such as ellagic acid [16]. We have previously reported the protective effects of chestnut inner shell extract (CIE) against allergic asthma [17]. Furthermore, ellagic acid has been reported to have diverse pharmaceutical properties, especially in various pulmonary diseases such as fibrosis, inflammation, and emphysema [18,19,20,21,22]. Based on these results, we hypothesized that CIE would be a potential protective agent against emphysema in a mouse model of COPD induced by cigarette smoke condensate (CSC) instillation.

To investigate this hypothesis, we elucidated the efficacy of CIE on CSC-induced emphysema by measuring macrophage cell counts and performing histological analysis. Additionally, to explain the underlying mechanisms, we elucidated the efficacy of CIE on protease levels in CSC-stimulated H292 cells.

## 2. Materials and Methods

### 2.1. Chestnut Inner Shell Extract Preparation

#### 2.1.1. Chestnut Inner Shell Extraction

Dried chestnut inner shell powder was purchased from Jangmyung (Seoul, Republic of Korea). The powder (300 g) was soaked in 70% ethanol (6 L) and incubated at room temperature for 1 day. The day after, the upper layer was filtered and concentrated using a rotary evaporator at 40 °C (Heidolph, Schwabach, Germany). Then, the extract was freeze-dried; the extraction yield was 25.05%.

#### 2.1.2. Chestnut Inner Shell Extract Analysis

HPLC fingerprint analysis was conducted using HPLC with a UV detector (Agilent Technologies, Inc., Santa Clara, CA, USA), and separation was achieved using a C18 column (2.1 × 150 mm, 3.5 μm; Agilent) below 40 °C. The marker standard compounds, ellagic acid and gallic acid, were obtained from Sigma-Aldrich (CA, USA; >95% purity). Gradient elution was conducted with 0.1% formic acid in water as mobile phase A and acetonitrile as mobile phase B. The flow rate was 0.25 mL/min, and the gradient conditions were set as follows: 0–1 min, hold at 3% B; 1–2 min, increase to 15% B; 2–13 min, increase to 50% B; 13–20 min, increase to 100% B; 20–23 min, hold at 100% B; and 23–23.5 min, return to 3% B. The injection volume was 5 μL, and the detection wavelength was set at 280 nm.

### 2.2. In Vivo Experiments

#### 2.2.1. Experimental Animal Model

Male mice of C57BL/6 (6 weeks old, 20–24 g) were obtained from ORIENT BIO. (Daejeon, Republic of Korea) and used after quarantine and acclimatization for one week. Total animal experimental schedules were recognized by the Animal Care and Use Committee of Chungnam National University. The experimental mice were split into four groups: (1) the NC (vehicle intranasal instillation + vehicle oral (p.o.)) group; (2) CSC (CSC intranasal instillation, 50 mg/kg/week + vehicle p.o.) group; and (3) CSC + CIE100 and CSC + CIE300 (CSC intranasal instillation, 50 mg/kg/week + CIE p.o., 100 or 300 mg/kg/day, respectively) groups (*n* = 6 per group). CSC was provided from the Korea Institute of Toxicology (Jeongeup, Republic of Korea) and instilled via intranasal route during day 7 to day 35 at 7-day intervals for 1 month (total of 4 times) at 50 mg/kg in 30 μL vehicle. CIE was dissolved in vehicle and administered to mice by p.o. from one week prior to first CSC instillation to day 35 at 100 or 300 mg/kg/day. The mice were euthanized, and tracheostomy was performed 1 day after the last CSC instillation.

#### 2.2.2. Analysis of Macrophage Count in Bronchoalveolar Lavage Fluid (BALF)

To obtain BALF, the lungs of mice were irrigated with phosphate-buffered saline (PBS; 4 °C, total volume: 1.4 mL). After centrifugal processing, the pellet of BALF was isolated for cell counting. To count macrophage cells, resuspended BALF was attached on slides utilizing a cytospin (Hanil Science Industrial, Seoul, Republic of Korea). The cells stuck to the slides were processed with the Diff-Quik^®^ staining procedure (Sysmex Corporation, Kobe, Japan).

#### 2.2.3. Histological Analysis

Lung tissue was fixed and embedded according to general procedure and sectioned to 4 μm thickness. The sections were dyed with hematoxylin and eosin reagents. To evaluate emphysematous lesions, all observations were randomly selected and evaluated manually using a light microscope with ×10 and ×20 objective lenses and a ×100 immersion lens. The mean linear intercept length (MLI) was calculated manually according to a previous study [23].

#### 2.2.4. Immunohistochemistry (IHC) and Gelatin Zymography

For IHC of MMP-9, lung sections were processed using an IHC kit in accordance with the general protocol (Vector ABC Elite Kit, Vector Laboratories, Mowry Ave Newark, CA, USA). After incubation with anti-MMP-9 (1:200 dilution; Abcam, Cambridge, UK) at ordinary temperature for 2 h, the slides were rinsed with PBS mixed with 0.1% Tween 20 and developed with biotinylated anti-rabbit IgG (30 min, ordinary temperature). After washing again, the sections were stained with 3,3-diaminobenzidine chromogen and hematoxylin for visualizing. Each slide was examined in a blinded manner under a light microscope (Leica). Ten non-overlapping fields per slide were randomly obtained, and image analysis was performed for quantization using an image analyzer (IMT i-Solution software, Houston, TX, USA).

To confirm MMP-9 activity, lung samples of each group were analyzed by zymography. After loading into gelatin gel, the protein was incubated with developing buffer (1 M Tris-HCl pH 7.5 with 10 mM CaCl_2_) and stained with 45% methanol/10% acetic acid containing Coomassie brilliant blue. After washing with de-staining buffer I and II, white bands in the gel were used to identify proteolysis caused by MMP-9.

#### 2.2.5. Immunoblotting

Lung tissues were grinded utilizing an auto homogenizer (Hangzhou Allsheng Instruments Co., Hangzhou, China) with lysis buffer (Sigma-Aldrich, St. Louis, MO, USA) containing a protease inhibitor cocktail (Sigma-Aldrich) and a phosphatase inhibitor cocktail (Sigma-Aldrich). After determining the protein concentration of each sample with a BCA reagent kit (Thermo Scientific, Waltham, MA, USA), equal quantities of protein (30 μg) were loaded and transferred according to general procedure. The membrane was incubated with 5% BSA for 1 h at ordinary temperature for blocking, and then incubated overnight (4 °C) with the first antibody. The following antibodies were used: total phosphatidylinositol 3-kinase (t-PI3K; 1:1000 dilution; Abcam), phosphor (p)-PI3K (1:1000 dilution; Abcam), t-protein kinase B (t-AKT; 1:1000 dilution; Abcam), p-AKT (1:1000 dilution; Abcam), t-nuclear factor kappa B (t-NF-κB; 1:1000 dilution; Abcam), p-NF-κB (1:1000 dilution; Abcam), MMP-9 (1:1000 dilution; Novus Biologicals, Littleton, CO), and β-actin (1:4000 dilution; Cell Signaling Technology, Danvers, MA, USA). The membrane was washed with PBS mixed with 0.1% Tween 20 and incubated with a horseradish-peroxidase-conjugated secondary antibody at 1:5000 dilution (AbFrontier Co., Seoul, Republic of Korea) for 2 h at room temperature. After washing again, development was performed using a chemiluminescent solution kit (Biomax Inc., Seoul, Republic of Korea). Relative expression values were determined using a ChemiDoc (Bio-Rad Laboratories, Hercules, CA, USA).

### 2.3. In Vitro Experiments

#### 2.3.1. Cell Culture and Cell Viability

NCI-H292, a human airway epithelial cell line (American Type Culture Collection, Manassas, VA, USA), was grown in indicated medium (RPMI 1640, 10% fetal bovine serum, 1% penicillin–streptomycin) at 37 °C with 5% CO_2_. Cell viability was confirmed using an EZ-Cytox (DoGenBio Co., Seoul, Korea) in the presence of CSC and CIE. NCI-H292 cells were seeded at a density of 2.5 × 10^4^ cells/well in 96-well plates and incubated overnight. The cells were incubated with various concentrations of CSC (0, 6.25, 12.5, 25, 50, and 100 μg/mL) and CIE (0, 5, 10, 20, 40, and 80 μg/mL) using twofold serial dilution. After 24 h of treatment, 10 μL of an EZ-Cytox solution was added to each well and the cells were cultured at 37 °C with 5% CO_2_ for 1 h. Cell viability was measured as absorbance at 450 nm utilizing an enzyme-linked immunosorbent assay reader (Bio-Rad Laboratories) and compared with the measured value of untreated cells.

#### 2.3.2. Analysis of mRNA Expression Levels

NCI-H292 cells were seeded on a six-well plate at a concentration of 6 × 10^5^ cells/well and incubated for 24 h; then, CIE was added to the wells at non-toxic concentrations (5, 10, and 20 μg/mL) according to the measured cell viability results. After 15 h of CIE pretreatment, the cells were cultured with CSC at 100 μg/mL for 4 h. Quantitative reverse-transcription polymerase chain reaction (qRT-PCR) was operated to analyze the mRNA expression levels of cytokines. Total RNA was collected using a total RNA extraction kit (Samjung Bioscience, Daejeon, Republic of Korea) according to general protocol, and the collected RNA was reverse-transcribed into cDNA using a cDNA synthesis kit (Samjung Bioscience). qRT-PCR was performed using specific forward and reverse primers (interleukin (IL)-1β, IL-6, and tumor necrosis factor (TNF)-α: forward).

#### 2.3.3. Immunoblotting

NCI-H292 cells were seeded on a six-well plate at a concentration of 6 × 10^5^ cells/well and incubated for 24 h; then, CIE was added to the wells at non-toxic concentrations (5, 10, and 20 μg/mL) according to the measured cell viability results. After 15 h of CIE pretreatment, the cells were cultured with CSC at 100 μg/mL for various times according to each protein expression value. Thereafter, the cells were rinsed twice with PBS (4 °C) and lysed by adding RIPA buffer (Sigma-Aldrich) containing protease and phosphatase inhibitors. The immunoblotting procedure and antibodies used are described above. Relative protein expression was confirmed using a ChemiDoc (Bio-Rad Laboratories).

### 2.4. Statistical Analysis

The results were expressed as means ± standard deviations (SD), and all statistical comparisons were performed via one-way analysis of variance followed by Tukey’s post hoc honestly significant difference test. Statistical significance between the treatment groups and NC group was determined using GraphPad InStat v3.0 (GraphPad Software, Inc., La Jolla, CA, USA). Statistical significance was set at *p* < 0.05 and <0.01.

## 3. Results

### 3.1. CIE Contents

The marker compounds in CIE were analyzed using HPLC-UV (Figure 1). The CIE peak was confirmed by comparing the retention time and consistency of the reference peak with those of the standard compounds. The retention time for gallic acid and ellagic acid were about 3.8 min and 10.5 min, respectively, and their contents in CIE were approximately 0.04% and 0.17%.

### 3.2. In Vivo Experiments

#### 3.2.1. Effects of CIE on Macrophage Counts in BALF from CSC-Instilled Mice

As shown in Figure 2A, the CSC-exposed mice showed a significant increase in the number of macrophages in BALF compared with the NC group. In contrast, CIE-treated groups (100 and 300 mg/kg/day) had lower numbers of macrophages in comparison to the CSC-exposed group, while the numbers of neutrophils showed no statistical significance between CSC group with CIE groups (Appendix A).

#### 3.2.2. Effects of CIE on Emphysematous Lesions in CSC-Instilled Mice

Histological analysis revealed that CSC-exposed mice exhibited extensive emphysematous alveolar lesions, as indicated by the enlarged air space in the pulmonary alveolar (Figure 2B). However, the CIE-treated mice showed significantly fewer emphysematous lesions compared with the CSC-exposed mice, which results were revealed by MLI.

#### 3.2.3. Effects of CIE on MMP-9 Expression in Lung Tissues of CSC-Instilled Mice

We also investigated the efficacy of CIE on expression of MMP-9 in CSC-exposed mice. These mice showed higher MMP-9 positivity in lung tissues, as revealed by IHC, compared with the NC group (Figure 3A and Appendix A). However, CIE-treated mice (100 and 300 mg/kg/day) showed lower MMP-9 positivity in lung tissues compared with CSC-exposed mice, and these decreases were more noticeable in the CSC + CIE300 group. Similar to MMP-9 positivity, MMP-9 expression, which was evaluated by Western blotting, significantly reduced in the CIE-treated groups compared to the CSC group (Figure 3B). The activity of MMP-9, confirmed through zymography, also showed a trend similar to the results of IHC and Western blotting. The CSC group showed a higher activity of MMP-9 than the NC group, whereas the CIE-treated groups exhibited lower activity in a dose-dependent manner (Figure 3C).

#### 3.2.4. Effects of CIE on PI3K/AKT/NF-κB Pathway in Lung Tissues of CSC-Instilled Mice

To elucidate the underlying mechanism of upregulation of protease expression upon CSC instillation, immunoblotting was performed to analyze the phosphorylation of PI3K/AKT/NF-κB. As shown in Figure 4 and Appendix A, the CSC-exposed group expressed a marked elevation in the phosphorylation of PI3K compared to the NC group. In contrast, CIE-treated groups (100 and 300 mg/kg/day) showed a marked reduction in the phosphorylation of PI3K compared to the CSC-instilled mice. Similarly, compared to the NC group, the CSC-treated group expressed marked elevations in the phosphorylation of AKT and NF-κB. However, the increase induced by CSC instillation significantly decreased upon CIE treatment.

### 3.3. In Vitro Experiments

#### 3.3.1. Effects of CIE on Cell Viability and mRNA Expression of Cytokines in NCI-H292 Cells Stimulated with CSC

In H292 cells, the non-toxic effects of CSC and CIE were observed at concentrations of up to 100 and 20 μg/mL, respectively (Figure 5A,B). Therefore, we used CIE concentrations of up to 20 μg/mL and CSC concentrations of up to 100 μg/mL in the experiments. The mRNA expression levels of IL-1β were higher in CSC-stimulated H292 cells than in untreated cells (Figure 5C). On the contrary, the levels of IL-1β tended to decrease with CIE treatment. In line with these results, the mRNA levels of IL-6 and TNF-α showed a similar trend (Figure 5D,E). The IL-6 and TNF-α mRNA levels, which were upregulated by CSC treatment, significantly reduced upon CIE treatment.

#### 3.3.2. Effects of CIE on Expression of PI3K/AKT NF-κB Pathway in NCI-H292 Cells Stimulated with CSC

CSC-stimulated H292 cells exhibited higher levels of PI3K phosphorylation than those in untreated cells (Figure 6A and Appendix A). However, cells treated with CIE showed lower levels of PI3K phosphorylation compared to CSC-treated cells. Similar to the results of the in vivo experiments, increased phosphorylation of PI3K was followed by upregulated phosphorylation of AKT and NF-κB (Figure 6A, Figure 7A, and Appendix A). The phosphorylation levels were higher in CSC-stimulated cells than in untreated cells, whereas they were lower after CIE treatment. The decrease in protein phosphorylation was concentration-dependent.

#### 3.3.3. Effects of CIE on MMP-9 Expression in NCI-H292 Cells Stimulated with CSC

As shown in Figure 7A, CSC stimulation increased MMP-9 expression in H292 cells. The CIE treatment groups showed lower MMP-9 expression compared with CSC-stimulated cells in a concentration-dependent manner. These results are consistent with the expression of MMP-9 confirmed in vivo and the expression of proteins thought to be upstream signals, verified by in vivo and in vitro experiments.

## 4. Discussion

Cigarette smoking is a major risk agent in the COPD progression, which is characterized by destructive enforcement of airspaces [24]. These progressions, called emphysema, stem from the accumulation of macrophages, which results in protease–antiprotease imbalance and eventually causes the destruction of normal alveolar architecture [25]. In the present study, we evaluated the impact of macrophages on emphysema progression using a CSC-induced COPD mouse model. In addition, we studied the preventive effects of CIE on CSC-induced emphysema progression in vivo and in vitro, focusing on MMP-9 associated pathways. CIE significantly decreased the number of macrophages in BALF and the number of emphysematous lesions induced by CSC instillation. Additionally, CIE suppressed the PI3K/AKT/NF-κB pathway and thereby downregulated proteolytic enzyme MMP-9 in CSC-instilled mice and CSC-treated H292 cells.

Macrophages are crucial innate immune system effector cells and are primarily located where the organisms react with the environment [7]. Considering the anatomical structure of the lungs, which are inevitably exposed to external pollutants, macrophages are closely related to diseases caused by external substances, such as CS [26]. Macrophages are implicated in COPD pathogenesis, and macrophage infiltration in lung tissues correlates with COPD severity [27,28]. As the number of macrophages in patients’ BALF, lung, and sputum samples have been observed in various COPD clinical cases, increased macrophage numbers in the parenchyma are associated with emphysema severity [29,30,31,32]. In particular, it has been reported that the population of macrophages was increased in the alveolar wall where destruction and emphysema occurred [33]. These results imply that macrophages have a potentially important part in the progression of emphysema. In this study, as the dosage of CIE treatment increased, the accumulation of macrophage cells decreased in the BALF of CSC-instilled mice. The effect of CIE treatment, which reduced the number of macrophages, was consistent with the attenuation of emphysematous lesions in lung tissues. Our results show that CIE suppresses emphysema by decreasing the macrophage population.

The alveolar macrophages of the lungs are frequently exposed to airborne agents and cause activation, phagocytosis, and respiratory bursts by interacting with environmental pathogenic particles [7]. These processes result in protease–antiprotease imbalance, which is one of the established mechanisms underlying COPD pathogenesis related to macrophages [8]. When protease–antiprotease imbalance progresses, tissue destruction and incorrect repair occur in the lungs, resulting in emphysema [34]. Previous studies have shown that airflow obstruction becomes more severe as the number of macrophages increases, the main source of proteases in the lungs [35,36,37]. Among several proteases, MMP-9 is known to have an important part in the progression of respiratory diseases, including COPD [38]. It is a primary matrix decomposition enzyme that catalyzes the cleavage of denatured collagen and basal membrane components [39]. According to Zheng et al. [40], overexpression of MMP-9 promotes the destruction of lung tissue, leading to emphysema. Furthermore, in clinical trials, high levels of MMP-9 have been observed in the lungs of COPD patients [41,42]. In the present study, CIE treatment reduced the elevated expression and activity of MMP-9 induced by CSC instillation, as revealed by IHC, Western blotting, and zymography. The efficacy of CIE on MMP-9 expression was confirmed using CSC-treated H292 cells. Collectively, the preventive effects of CIE against alveolar destruction are closely related to the suppression of MMP-9 expression.

The PI3K/AKT pathway is an intracellular signaling pathway associated with various cellular responses, including cell growth, migration, proliferation, and apoptosis [43,44]. It has been reported that phosphorylation of PI3K/AKT plays a crucial role in COPD progression [45,46]. According to Stark et al. [47], PI3K activates AKT phosphorylation as an upstream regulator of AKT in response to CS. AKT then regulates the signaling pathways related to the destructive pathogenesis of COPD. AKT induces NF-κB phosphorylation to adjust macrophage recruitment [48]. Wang et al. [46] reported that the PI3K/AKT/NF-κB signaling pathway plays a crucial role in lung ischemia–reperfusion injury, and [49] reported that the levels of activated NF-κB in COPD patients were significantly higher than those in normal individuals. Furthermore, the regulation of MMP-9 depends on the binding of NF-κB to the regulatory element in the MMP-9 gene, which results in the upregulation of MMP-9 enzymatic activity [50,51]. In various reports on respiratory diseases, the relationship between NF-κB and MMP-9 has been well demonstrated [52,53,54]. In the present study, CIE treatment significantly reduced the phosphorylation of the PI3K/AKT/NF-κB pathway in CSC-instilled mice. These results line up with those of CSC-stimulated H292 cells with/without CIE treatment. On the basis of previous studies and our results, the therapeutic effects of CIE against CSC-induced emphysema may be closely related to the inhibition of the PI3K/AKT/NF-κB signaling pathway.

The protective effects of CIE are attributed to its ellagic acid content. Ellagic acid, a polyphenolic compound, exhibits antioxidant and anti-inflammatory activity [55,56]. Previous studies have also demonstrated that ellagic acid decreases the number of inflammatory cells and reduces inflammatory cell infiltration into lung tissues through NF-κB signaling in LPS-induced mice [19,57]. In vitro experiments have shown that ellagic acid inactivates metalloproteinases such as MMP-9 [58]. These reports provide evidence that ellagic acid has potent therapeutic effects on lung diseases. This hypothesis was verified in our study that CIE, which contains ellagic acid, inhibited the PI3K/AKT/NF-κB/MMP-9 pathway in CSC-stimulated H292 cells. However, the analysis of the components in CIE has not been fully elucidated, and the identification of the leading active component is an essential step for drug development that can offer new therapeutic options for clinical use.

## 5. Conclusions

In summary, CIE reduced the macrophage population and severity of emphysematous lesions in CSC-exposed mice. These effects are closely related to the inhibition of protease–antiprotease imbalance, represented by the downregulation of MMP-9 expression through the PI3K/AKT/NF-κB signaling pathway, as found via in vitro experiments. Although further studies are needed to investigate the direct effects of macrophage on emphysema progression using specific protein knock-out mice or short interfering RNA methods, our results imply that CIE may be an effective supplemental agent for the alleviation of emphysema progression in COPD patients.

## Figures and Tables

**Figure 1 nutrients-15-00253-f001:**
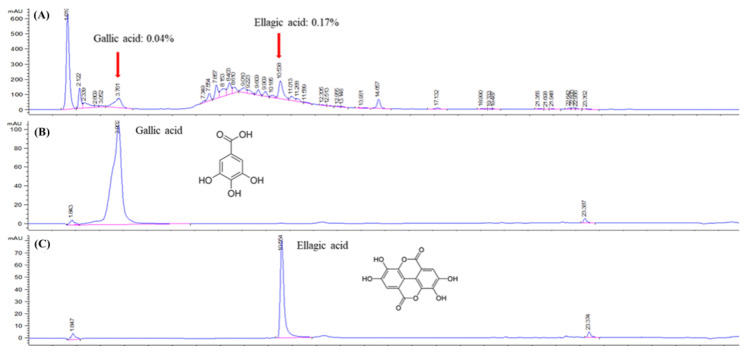
HPLC fingerprinting of chestnut inner shell (**A**) and chromatogram of reference standards (**B**,**C**).

**Figure 2 nutrients-15-00253-f002:**
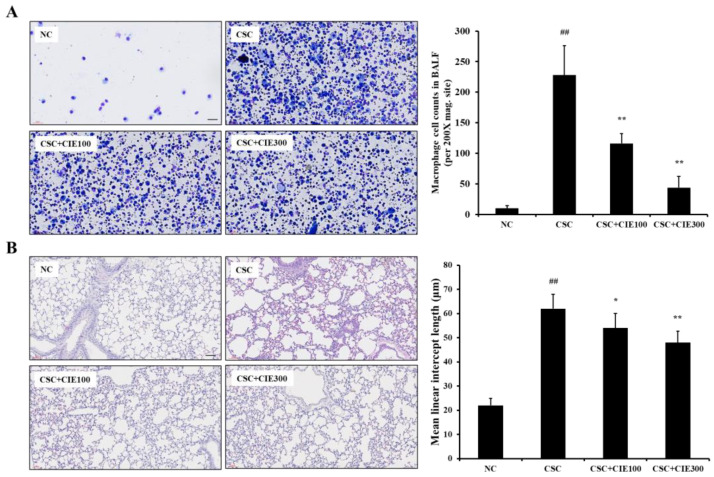
CIE reduces the elevated (**A**) macrophage cell counts in BALF and alleviates the (**B**) emphysematous lesions in lung tissues. NC; mice instilled with vehicle + vehicle p.o., CSC; mice instilled with CSC + vehicle p.o., CSC + CIE100 and CSC + CIE300; mice instilled with CSC + CIE p.o., 100 or 300 mg/kg/day, respectively. Values: means ± SD (*n* = 6). Significance: ## *p* < 0.01 vs. NC; *, ** *p* < 0.05 and 0.01 vs. CSC, respectively. Bar = 60 μm.

**Figure 3 nutrients-15-00253-f003:**
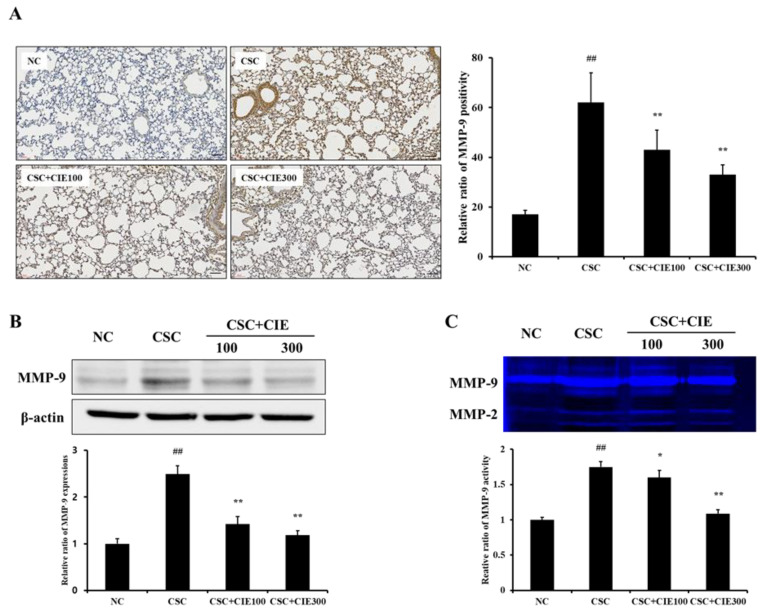
CIE reduces the elevated (**A**) MMP-9 positivity, (**B**) MMP-9 expression, and (**C**) MMP-9 activity in lung tissues from mice, as revealed by IHC, Western blot, and zymography, respectively. NC; mice instilled with vehicle + vehicle p.o., CSC; mice instilled with CSC + vehicle p.o., CSC + CIE100 and CSC + CIE300; mice instilled with CSC + CIE p.o., 100 or 300 mg/kg/day, respectively. Values: means ± SD (*n* = 6). Significance: ## *p* < 0.01 vs. NC; *, ** *p* < 0.05 and 0.01 vs. CSC, respectively. Bar = 60 μm.

**Figure 4 nutrients-15-00253-f004:**
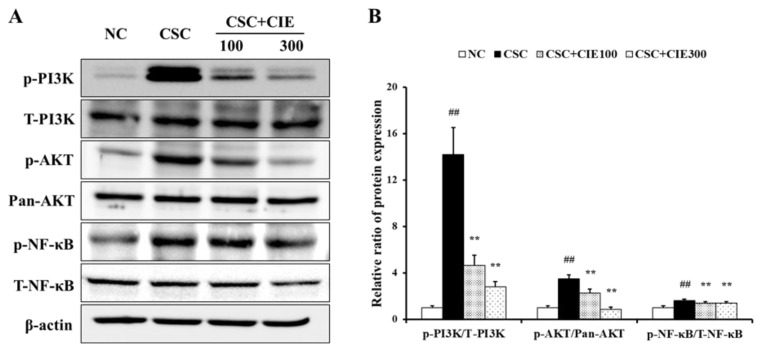
CIE reduces the phosphorylation of (**A**) PI3K, AKT, NF-κB, of which (**B**) densitometric values were investigated utilizing ChemiDoc. NC; mice instilled with vehicle + vehicle p.o., CSC; mice instilled with CSC + vehicle p.o., CSC + CIE100 and CSC + CIE300; mice instilled with CSC + CIE p.o., 100 or 300 mg/kg/day, respectively. Values: means ± SD (*n* = 6). Significance: ## *p* < 0.01 vs. NC; ** *p* < 0.01 vs. CSC.

**Figure 5 nutrients-15-00253-f005:**
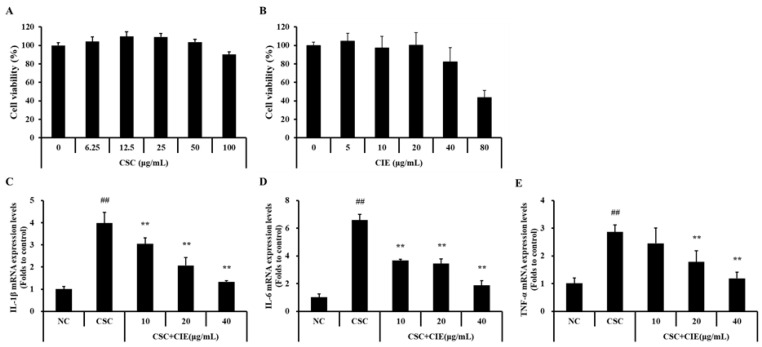
The doses of (**A**) CSC and (**B**) CIE used in the experiments were determined according to the results of cell viability tests on NCI-H292 cells. CIE reduces the up-regulated levels of (**C**) IL-1β, (**D**) IL-6, and (**E**) TNF-α in the CSC-stimulated cells. NC; non-treated cells, CSC; stimulation of CSC, CIE10, 20, and 40; stimulation of CSC + treatment of 10, 20, and 40 μg/mL of CIE. Values: means ± SD (*n* = 3). Significance: ## *p* < 0.01 vs. NC; ** *p* < 0.01 vs. CSC.

**Figure 6 nutrients-15-00253-f006:**
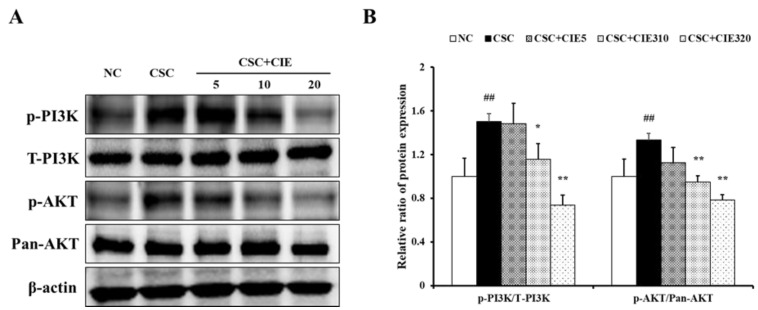
CIE reduces the elevated phosphorylation of (**A**) PI3K and AKT. (**B**) The densitometric values were determined using ChemiDoc. NC; non-treated cells, CSC; stimulation of CSC, CIE10, 20, and 40; stimulation of CSC + treatment of 10, 20, and 40 μg/mL of CIE. Values: means ± SD (*n* = 3). Significance: ## *p* < 0.01 vs. NC; *, ** *p* < 0.05 and 0.01 vs. CSC, respectively.

**Figure 7 nutrients-15-00253-f007:**
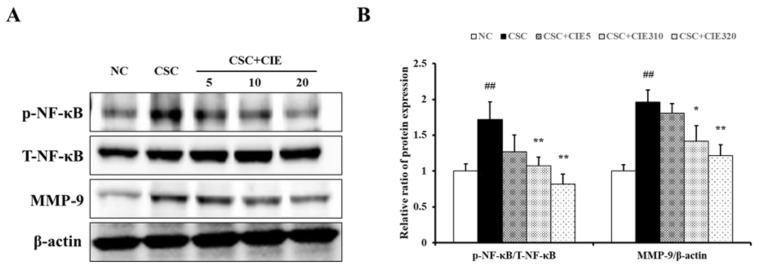
CIE reduces the elevated (**A**) phosphorylation of NF-κB and expression of MMP-9. (**B**) The densitometric values were determined using ChemiDoc. NC; non-treated cells, CSC; stimulation of CSC, CIE10, 20, and 40; stimulation of CSC + treatment of 10, 20, and 40 μg/mL of CIE. Values: means ± SD (*n* = 3). Significance: ## *p* < 0.01 vs. NC; *, ** *p* < 0.05 and 0.01 vs. CSC, respectively.

## Data Availability

Data are contained within the article.

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
