# Peer review of "Protective Effects of Chestnut (Castanea crenata) Inner Shell Extract in Macrophage-Driven Emphysematous Lesion Induced by Cigarette Smoke Condensate"

_nutrients, 2023, doi:10.3390/nu15020253_

Round 1
Reviewer 1 Report
Dear all, first I would like to congratulate the authors for their work. However, I suggest some revisions to be made.
As a first comment, I believe that the methodological description of both the instillation of the CSC and the CIE is not well described. On line 90 for example, the authors say that the CSC was administered every 7 days for 1 month. Is that a total of 4 instillations? I think the information on the total number of instillations should be clearer. As in line 92, it also does not say the frequency of CIE administration and whether it was before, during or after CSC administration. This is repeated on line 152 in the in vitro part; the timing of CIE administration will only be mentioned in the mRNA analysis part.
Regarding the results, figure 2A states that there is a dose-dependent effect on the decrease in the number of macrophages in the BAL when the mouse is treated with CIE. However, despite the appearance of a downward trend in the treatment with 300mg/kg/day, the difference is the same (p<0.01) in the treatment with 100mg/kg/day in relation to the CSC group. I think it is inappropriate to say that the effect is dose-dependent. Also, the image is not representative of the graph. In the NC group there are much less cells than in the CSC+CIE300 group and yet the values are very similar in the graph (around 50).
Analyzing the WB images in Figure 3B and comparing them with the supplementary images, I have a doubt regarding the height of the band. According to the authors, the molecular weight of MMP-9 is 82 or 63kDa; however, the band considered for MMP-9 is slightly below the height of 55kDa in the 3 replicates. I believe this WB could be revised.
Once again, there is a statement of dose-dependent action in figures 5C, 5D and 5E. I agree that there is a trend, but there is no significant difference between the CSC+CIE 10, 20 and 40ug/ml groups.
Regarding the discussion and conclusion of the paper, I believe it is too precipitate to say that the role of macrophages in emphysema was investigated and that the CIE acted through macrophages so that lung tissue repair occurred, since macrophages were analyzed only by differential count from BAL and indirectly by expression of MMP-9. I believe that to make this statement some experiments with purified or differentiated macrophages should be done, or the use of knockout mice, or some other specific experiment that shows a direct effect and not just a causal one. Furthermore, the statement that the effect of CIE on the lung is due to ellagic acid is also premature since its amount seen in figure 1 is small (0.17%), and to state this the authors should have done at least an experiment with isolated ellagic acid (in vitro and/or in vivo) to, again, show the direct effect and not by deduction.
I believe that, with some improvements, the results would be more robust and the paper would be even more interesting for the scientific community.
Author Response
Reviewer #1: Dear all, first I would like to congratulate the authors for their work. However, I suggest some revisions to be made.
As a first comment, I believe that the methodological description of both the instillation of the CSC and the CIE is not well described. On line 90 for example, the authors say that the CSC was administered every 7 days for 1 month. Is that a total of 4 instillations? I think the information on the total number of instillations should be clearer. As in line 92, it also does not say the frequency of CIE administration and whether it was before, during or after CSC administration. This is repeated on line 152 in the in vitro part; the timing of CIE administration will only be mentioned in the mRNA analysis part.
- We appreciated your comments. We supplemented our manuscript by providing experimental schedule in materials and methods section of in vivo and in vitro (immunoblotting part) experiments parts.
Regarding the results, figure 2A states that there is a dose-dependent effect on the decrease in the number of macrophages in the BAL when the mouse is treated with CIE. However, despite the appearance of a downward trend in the treatment with 300mg/kg/day, the difference is the same (p<0.01) in the treatment with 100mg/kg/day in relation to the CSC group. I think it is inappropriate to say that the effect is dose-dependent. Also, the image is not representative of the graph. In the NC group there are much less cells than in the CSC+CIE300 group and yet the values are very similar in the graph (around 50).
- We appreciated your comments. As you pointed out, we modified the results by deleting the term “dose-dependent” for better interpretation. Also, we corrected graph of figure 2A that were misrepresented due to the incorrectly calculated NC group at a factor of 4 when scaling.
Analyzing the WB images in Figure 3B and comparing them with the supplementary images, I have a doubt regarding the height of the band. According to the authors, the molecular weight of MMP-9 is 82 or 63kDa; however, the band considered for MMP-9 is slightly below the height of 55kDa in the 3 replicates. I believe this WB could be revised.
- Thanks for your kind comments. It is thought that there are various reasons differences from predicted band size from real band image size. According to our experimental experience so far, the 5 major reasons are as follows: inadequacy of tissue sampling process, buffer composition, pH, antibody purity, and blocking. These reasons may cause not only the generation of an unspecific band but also the splitting of the one target band into several strands at the expected molecular weight. Also, while there are antibodies that show these phenomena well, there are antibodies that show only the target band well. In addition to the above reasons, the band expressed as MMP-9 in the present study is judged as a positive band in that the band did not appear at the expected height and the band image of the strong signal appeared at the expected band position. We hope that our answer will be an appropriate response to your comments.
Once again, there is a statement of dose-dependent action in figures 5C, 5D and 5E. I agree that there is a trend, but there is no significant difference between the CSC+CIE 10, 20 and 40ug/ml groups.
- We appreciated your comments. As you pointed out, we modified the results by deleting the term “dose-dependent” for better interpretation.
Regarding the discussion and conclusion of the paper, I believe it is too precipitate to say that the role of macrophages in emphysema was investigated and that the CIE acted through macrophages so that lung tissue repair occurred, since macrophages were analyzed only by differential count from BAL and indirectly by expression of MMP-9. I believe that to make this statement some experiments with purified or differentiated macrophages should be done, or the use of knockout mice, or some other specific experiment that shows a direct effect and not just a causal one.
- We appreciated your comments. We fully agree that what you pointed out is a limitation of our experimental results. To compensate for this, we agree that the experiments you pointed out are necessary, but we hope you understand that it is not easy to perform within the current deadline. An explanation to supplement these limitations was written in introduction and discussion section using sufficient references, additionally, we inserted the limitations of our experiment and the need for further study at the end of the discussion.
Furthermore, the statement that the effect of CIE on the lung is due to ellagic acid is also premature since its amount seen in figure 1 is small (0.17%), and to state this the authors should have done at least an experiment with isolated ellagic acid (in vitro and/or in vivo) to, again, show the direct effect and not by deduction.
- We appreciated your comments. It is too speculative to determine ellagic acid as leading active component for anti-emphysema effect of CIE. The present ellagic acid was analyzed as a marker component to secure quality of CIE used entire study in part. Natural products have various active ingredients, and they interact with each other to produce various biological effects and the effect might be derived from the synergistic integration of multiple components. Chestnut inner shell extracts have abundant natural polyphenols with health-promoting effects, and predominantly active components are tannins such as gallic acid and ellagic acid. Previously, many studies reported beneficial biological effect of ellagic acid on airway discomforts including, emphysema, we thus speculate that ellagic acid might take a role in part on anti-emphysema effect of CIE. In deed, the analysis of the components in CIE has not been fully elucidated, and the identification of the leading active component is an essential step for drug development that can offer new therapeutic options for clinical use. We are currently preparing peak-by-peak fractions to identify the active ingredients in CIE and we will subsequently compare the effect of component from CIE in the further study. We added these limitations briefly in end of discussion section of present manuscript.
References
Aslan, A.; Hussein, Y.T.; Gok, O.; Beyaz, S.; Erman, O.; Baspinar, S. Ellagic acid ameliorates lung damage in rats via modulating antioxidant activities, inhibitory effects on inflammatory mediators and apoptosis-inducing activities. Environ Sci Pollut Res Int. 2020, 27(7), 7526-7537.
Mansouri, Z.; Dianat, M.; Radan, M.; Badavi, M. Ellagic Acid Ameliorates Lung Inflammation and Heart Oxidative Stress in Elastase-Induced Emphysema Model in Rat. Inflammation. 2020, 43(3), 1143-1156.
Alves Cde, F.; Angeli, G.N.; Favarin, D.C.; de Andrade, E.L.; Chica, J.E.; Faccioli, L.H.; da Silva, P.R.; Rogerio Ade, P. The effects of proresolution of ellagic acid in an experimental model of allergic airway inflammation. Mediators Inflamm. 2013, 2013, 863198.

Reviewer 2 Report
The paper has interesting results and it is an important topic to be avaluated. However, some considerations must be done:
Neutrophils has an important role in the course of emphysema. From that, others questions are coming:
1. Why just macrophages were counted?
2. Once you haven’t evaluated the effect of CIE in neutrophils number, how do you know if the effect of CIE in emphysema is not by neutrophil modulation?
If you want focus on macrophage activity/modulation you should include it on the title to make clearer your idea
NCI-H292, a human airway epithelial cell line (line 142). Once you want investigate the macrophages role, why haven’t use macrophages? You could have collected peritoneal macrophages, lung macrophages from BALF… How can you explain to use other cells? Is it make sense?
Author Response
Reviewer #2: The paper has interesting results and it is an important topic to be avaluated. However, some considerations must be done:
Neutrophils has an important role in the course of emphysema. From that, others questions are coming:
- Why just macrophages were counted?
- We appreciated your comments. As a result of BALF cell count in our experiment, it was observed that macrophages increased with CSC and decreased with CIE. Although, compared to those results, neutrophils did not show evident differences in all groups, we added neutrophil counts data to supplemental figure 5.
- Once you haven’t evaluated the effect of CIE in neutrophils number, how do you know if the effect of CIE in emphysema is not by neutrophil modulation?
- From the BALF cell count results, it was confirmed that macrophage and neutrophil are mediator of CSC-induced damage. However, according to supplemental figure 5, CIE did not show differences between CSC and CIE groups. Thus, we focused macrophage modulation affected by CIE administration. As explained in present manuscript, COPD is a complex disease, and various mediators are involved in the pathogenesis of COPD. Smoking specifically modulates alveolar macrophages to infiltrate lung tissue. When comparing alveolar macrophages of smokers and non-smokers, changes in alveolar macrophage physiology associated with the stress of cigarette smoking cause excessive inflammation, resulting in fibrosis in small airways, and destruction of lung parenchyma due to excessive protease release. The up-regulation of proteolytic enzymes by alveolar macrophage in response to smoking has been advanced as crucial in the pathogenesis of COPD, in particular in the alveolar destruction that characterizes emphysema.
References
Barnes, P. J. Alveolar macrophages as orchestrators of COPD. COPD: Journal of Chronic Obstructive Pulmonary Disease. 2004, 1(1), 59-70.
Heguy, A.; O'Connor, T.P.; Luettich, K.; Worgall, S.; Cieciuch, A.; Harvey, B.G.; Hackett, N.R.; Crystal, R.G. Gene expression profiling of human alveolar macrophages of phenotypically normal smokers and nonsmokers reveals a previously unrecognized subset of genes modulated by cigarette smoking. Journal of Molecular Medicine (Berline, Germany). 2006, 84(4), 318-28.
Yamasaki, K.; van Eeden, S. F. Lung macrophage phenotypes and functional responses: role in the pathogenesis of COPD. International journal of molecular sciences, 2018, 19(2), 582.
If you want focus on macrophage activity/modulation you should include it on the title to make clearer your idea.
- Thanks for your kind comments. As your suggestion, we changed the title to emphasize the role of macrophages
NCI-H292, a human airway epithelial cell line (line 142). Once you want investigate the macrophages role, why haven’t use macrophages? You could have collected peritoneal macrophages, lung macrophages from BALF… How can you explain to use other cells? Is it make sense?
- Thanks for your kind comments. NCI-H292 cells are used to determine if CIE have a direct protective effect against epithelial cell damage caused by CSC through signals identified in in vivo. Macrophage is an important mediator that bursts damage caused by CSC. As you mentioned, it is the best methods to co-culture the macrophage cell line and airway epithelial cell line for accurate results, but it has not been implemented in our laboratory. A future study is needed for this part.

Round 2
Reviewer 1 Report
Dear authors,
Thank you for answering my questions and considering my comments. I am satisfied with the answers and with the changes that could be made within the short deadline available. I am happy to hear that more analyzes are being done to investigate ellagic acid and other components of CIE. I look forward to seeing these results in the future. Once again congratulations on the work.